# Federated Learning for Predicting the Next Node in Action Flows

**Daniel Lopes**
Instituto Superior Técnico, Universidade de Lisboa
`daniel.f.lopes@tecnico.ulisboa.pt`

**João Nadkarni**
OutSystems
`joao.nadkarni@outsystems.com`

**Filipe Assunção**
OutSystems
`filipe.assuncao@outsystems.com`

**Miguel Lopes**
OutSystems
`miguel.gomes.lopes@outsystems.com`

**Luís Rodrigues**
INESC-ID, Instituto Superior Técnico, Universidade de Lisboa
`ler@tecnico.ulisboa.pt`

## Abstract

Federated learning is a machine learning approach that allows different clients to collaboratively train a common model without sharing their data sets. Since clients have different data and classify data differently, there is a trade-off between the generality of the common model and the personalization of the classification results. Current approaches rely on using a combination of a global model, common to all clients, and multiple local models, that support personalization. In this paper, we report the results of a study, where we have applied some of these approaches to a concrete use case, namely the *Service Studio* platform from OUTSYSTEMS, where Graph Neural Networks help programmers in the development of applications. Our results show that the amount of data points of each client affects the personalization strategy and that there is no optimal strategy that fits all clients.

## 1 Introduction

In this paper, we experimentally evaluate the performance of different personalization strategies when Federated Learning (FL) is used to improve the productivity of the users of the *Service Studio* platform developed by OUTSYSTEMS[1]. This platform, among other functionalities, allows the creation of action flows, that is, sequences of actions that define the application logic. Each action has one of several types, for example, it can be of the type "*assign*" (assigns a value to a variable), "*for*" (performs a cycle of actions), "*if*" (conditional execution), and so on. In this context, Machine Learning (ML) techniques are used to suggest to the user which action or actions can be added to a given action flow. Specifically, in this service the suggestions presented to the programmer are obtained from a model based in Graph Neural Networks (GNNs).

Our goal is to have a better understanding of how the model used to make recommendations to a given client should be constructed: considering the client's own data (local model), considering data from several clients, which can be aggregated centrally (centralized model), or combined in a federated

---

[1]OUTSYSTEMS specializes in the development of a platform that enables visual programming which is used by thousands of clients

Workshop on Federated Learning: Recent Advances and New Challenges, in Conjunction with NeurIPS 2022 (FL-NeurIPS'22). This workshop does not have official proceedings and this paper is non-archival.

way using one of different personalization strategies. Each of these approaches has its advantages and disadvantages when considering the quality of the recommendations provided.

Table 1: Comparison between model accuracies for different clients.

| | Number of Action Flows | Accuracy (%) Local Model | Accuracy (%) Centralized Model |
|---|---|---|---|
| **Client A** | 47,711 | 75.41 | 65.79 |
| **Client B** | 60 | 24.14 | 58.62 |

Table 1 highlights the advantages and disadvantages, in terms of the quality of the recommendations, of the usage of local models in relation to the usage of a single global model, calculated from the data of about 800 clients, resorting to two distinct clients. Client A has a long usage history of the platform, therefore, it already has a large data set. As such, we have enough data points to construct a local model which offers great accuracy and is specialized to its business model. For this client, the usage of a centralized model does not amount to any benefits, on the contrary, it leads to some loss of specialization. On the other hand, client B is relatively new to the platform, thus it has generated a small data set. This client clearly benefits from the usage of a centralized model. One way of avoiding this dichotomy is to resort to federated models with hybrid characteristics, that is, which are composed of a shared global part and a local part, which is specialized to each client. Currently, OUTSYSTEMS utilizes a model trained in a centralized environment. The goal of this study is to explore the viability of using an approach based on personalized federation to generate ML models for the OUTSYSTEMS use case which are able to maintain or improve the performance (measured in accuracy) of local and centralized models while ensuring client data privacy.

In the literature, there are several proposals of these personalized federated models which try to combine the advantages of both local and global models. However, these models have been proposed and evaluated in completely different domains such as natural language and image classification. As far as we are aware, there is no comparative study of these proposals of model personalization for models based on GNNs. Furthermore, previous work about federated models on GNNs focused only on the privacy of the client data[10]. Our work shows that for this use case, federated models can achieve performances very similar to the centralized model for smaller clients and to the local models for clients with a larger data set. Thus, federated approaches might be considered as an alternative to these models when it is necessary to use private and sensitive data.

## 2 Context

### 2.1 Action Flows

*Service Studio* platform allows its users to develop applications in a simple way, without coding, through the creation of action flows. The user simply needs to add actions of several types (each type has its corresponding functionality) and connect the actions in order to create a flow that represents the application logic.

Each action flow is represented by a directed graph. The nodes of the graph represent the actions and are connected by directed edges which represent the flow between two actions. Each node has its own attributes, for instance, the attribute "kind" which can be of one of several categories, such as, "*if*", "*assign*", "*for*", among others. The edges can also have attributes, for example, for a "*switch*" action, each edge contains an attribute that indicates the condition of the flow. In an action flow, there cannot be self-loops and there cannot be multiple edges between two nodes. Each flow also needs to have a node of the kind "*start*", where the flow begins, and it can end in nodes of the kind "*end*" or "*raise exception*", which cannot have outgoing edges.

Figure 1 shows an example of an OUTSYSTEMS Action Flow for splitting a string formatted in a given naming convention. The flow leverages a "switch" action to select the initial string naming convention format, either snake case (condition 1) or pascal case (condition 2), otherwise, it raises an exception. For the snake case, it first uses a "server action", which runs logic on the server, to

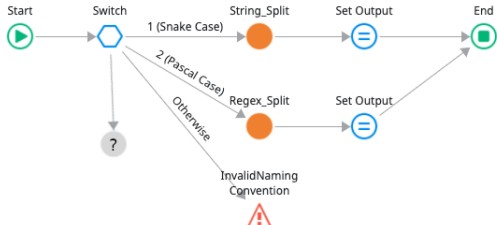

Figure 1: Action flow to split a string in a naming convention.

split the string by "_" and sets the output with an "assign" action. For example, for the input string "*federated_learning*" and the snake case naming convention, this flow outputs "*Federated Learning*". For the pascal case, first a "server action" is performed to split the string by capital letters and then the output is set. For example, from "*FederatedLearning*" we would obtain "*Federated Learning*".

## 2.2 Graph Neural Networks

As mentioned previously, each action flow is represented by a directed graph. GNNs allow the development of models capable of performing predictions on graphs. In the case of OUTSYSTEMS, the end goal of the model is to predict, from a finite set of kinds, the kind of one of the graph's nodes, marked as the target node. This task is classified as a node classification task. The prediction made is used by the platform to recommend possible next actions to the user. The Neural Network used by OUTSYSTEMS is based on the Message Passing architecture proposed by Battaglia et al. [1].

# 3 Studied Algorithms

## 3.1 Federated Learning

In its simplest form, the creation of ML models assumes that all the data of the clients is shared during the training phase. FL allows different clients to collaborate so as to construct a shared model without the need to share their private data, therefore preserving data privacy.

The most common approach to achieve FL consists in using a central server to orchestrate the coordination among clients. This architecture is described by Bonawitz et al. [3]. The protocol proceeds in rounds of communication where, in each round, the server selects a set of clients to participate. When the round starts, the server sends the parameters of the current global model to each participant. Afterwards, each participant independently trains the model received, using its own data set, obtaining a local model. The client then sends an update back to the server which reflects the changes that have been locally applied to the global model. Finally, the central server collects the updates from different clients, performs a weighted aggregation considering the size of each client's training data set, and uses the resulting global update to derive the new global model to be used in the following round. This aggregation method is defined as "Federated Averaging" (*FedAvg*) [8].

It is possible to define different FL categories, according to the way the data is partitioned, the way clients communicate and the scale of federation [5, 6, 11]. In this work, we consider a horizontally partitioned environment, with a centralized architecture and "cross-silo" federation, where the clients are the organizations using the *Service Studio* platform.

## 3.2 Personalized Federated Learning

FL introduces several challenges in different areas, including privacy of client data, robustness against attacks during the model training phase, communication efficiency and model performance. In this

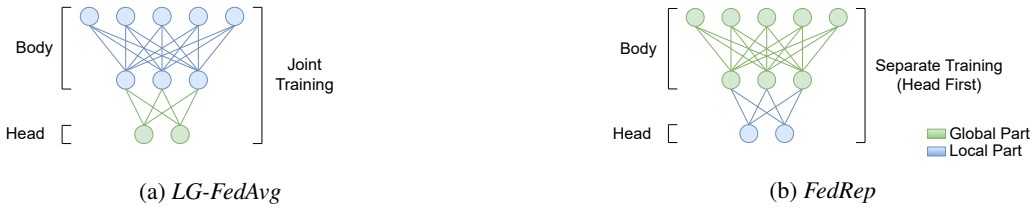

(a) *LG-FedAvg*  (b) *FedRep*

Figure 2: Models according to the algorithms *LG-FedAvg* and *FedRep*.

paper, we focus on the challenge of maximizing the performance of the model. This typically involves using mechanisms of model personalization [9]. Namely, the necessity of creating specific models for each client, which may include global components (which benefit from the contribution of all the clients) and specialized components (adjusted to the data set of each client).

Concretely, we focus on algorithms based on the parameter decoupling technique, which divides the model into two parts, the representation or "body" and the classifier or "head". The body is composed of the first layers of the model and is responsible for extracting the data features, the head is composed of the last layers of the model and is responsible for classifying the data from its features. Depending on the algorithm one of the parts is global and the other is local and specialized to each client. Clients can train both parts but the updates exchanged with the server correspond only to the global part. Two relevant examples of these approaches are "Local Global Federated Averaging" (*LG-FedAvg*) [7] and "Federated Representation Learning" (*FedRep*) [4]. We provide in the next paragraphs a brief description of these algorithms.

*LG-FedAvg* is an algorithm based on the parameter decoupling technique. In the specific case of this algorithm, the classifier is shared with the server and the representation is specialized for each client. Thus, this algorithm personalizes the body such that it can extract the features of the data for each client and shares the head in order to obtain a classifier that works for every client. Therefore, after receiving the head of the model from the server, the client associates its local body to obtain the local model which is, afterwards, trained in a joint way, that is, performing a sequence of local epochs and updating both the head and the body simultaneously. After training, the client sends to the server the updates referring to the head. Figure 2a illustrates the training procedure and the role of each model part.

*FedRep* is an algorithm which takes a different approach to the *LG-FedAvg* algorithm. The authors of the algorithm argue that results from centralized ML indicate that data shares a common representation of its features and that the heterogeneity resides in the classifications. Therefore, the representation is shared with the server and the classifier is specialized for each client. By sharing the body, the algorithm tries to obtain a global representation for all the clients while keeping the head local allows for the classifications to be specialized. Another difference between *FedRep* and *LG-FedAvg* resides in the way the training is performed. While in *LG-FedAvg* the body and the head are updated simultaneously and the for the same number of rounds, in *FedRep* the head is fully trained first and only afterwards is the body trained, furthermore, the number of training rounds between the head and the body may differ. Figure 2b illustrates the training procedure of the algorithm *FedRep*.

## 3.3 Comparison

Analyzing both algorithms, we can verify that as mentioned previously they take different approaches when it comes to the part of the model that is shared or personalized. In the case of *LG-FedAvg*, the objective is to allow for each client to have its own type of data, for instance, one client can have images while the other can have text. For that to happen, it needs the representation of the model to be personalized to each client so that it can extract the features of the data. The classifier is shared globally such that a classifier which works well for all clients can be obtained. However, a classifier that works for all clients does not mean it is the best for each one of the clients.

In terms of the *FedRep* algorithm, the objective is to find a representation which is common to every client, since the authors argue that centralized ML studies have shown that the data shares the same representation, only the classification is variable, for example, an image of a dog is represented

equally in two clients, however, one client can classify the dog as ugly while another can classify it as beautiful. Thus, the body is shared with the server and the head is personalized and kept locally.

When it comes to the training procedure, the algorithm *LG-FedAvg* trains both the head and the body simultaneously and for the same number of rounds, similarly to *FedAvg*. On the other hand, in the *FedRep* algorithm, the head is fully trained first and only then the body is trained with the head already specialized and both can be trained for a different number of rounds. The algorithm *FedRep* is, therefore, more flexible since it allows the parts of the model to be trained for a different number of rounds, which can be useful when we want to personalize the head further by performing more training rounds than the body, which is not possible in the *LG-FedAvg* algorithm.

# 4 Experimental Study

## 4.1 Experimental Setup

In order to evaluate each one of the federated algorithms, we have implemented both *LG-FedAvg* and *FedRep* algorithms using the Flower [2] framework (under the Apache License 2.0)[2]. We have used a proprietary dataset from OUTSYSTEMS consisting of the code developed by 881 clients. From this data set we selected 33 clients to be used in the study. Each client maintains the data relative to one organization which uses the *Service Studio* platform, that is, it keeps all the action flows of that organization. The data set splits were calculated in a per-client basis considering the following fractions: 0.8% for the train data set; 0.1% for the validation data set and; 0.1% for the test data set.

The 33 clients have been selected as follows: The clients were partitioned according to their number of action flows. The first partition includes all the clients with less than 64 flows and all the following partitions increase exponentially in size by a factor of 2, creating 11 partitions in total. Afterwards, 3 clients were randomly selected from each partition. Appendix A provides some statistics regarding the number of data points of the selected clients.

The evaluation was performed in the AWS cloud, where each client was run on a separate *t3.xlarge* instance. For the federated algorithms, 30 communication rounds were performed and for each one all the 33 clients participated in both training and testing, that is, there was no client selection since in the case of OUTSYSTEMS we assume no communication or hardware restrictions. In the case of the *FedAvg* and *LG-FedAvg* algorithms, a single local raining round was performed (the same number of local rounds used in the experimental evaluations of the *LG-FedAvg* [7] and *FedRep* [4] algorithms). For the *FedRep* algorithm, one local training round for the body (as used in the experimental evaluation of the *FedRep* [4] algorithm) and one for the head (since we wanted the head to have the same train conditions of the body) were performed. The local models were obtained using the data of each one of the 33 clients and the centralized model using the data of all the 33 clients in a single instance. Appendix C contains the used model parameters which were determined by an empirical evaluation previously performed by OUTSYSTEMS. Appendix D contains the experimental hyperparameters.

## 4.2 Model Accuracy

Since the data set is balanced (Appendix B contains the statistics on the distribution of classes for the 33 selected clients), the performance of the obtained models was evaluated using the accuracy of the models in each client's test data set, that is, the percentage of correct predictions over the total predictions. In the analysis of the results we split the clients into three groups, according to the percentiles of the number of data points of the 881 total clients: clients with a small number of data points (until 25% percentile, that is, 5300 data points); clients with an intermediate number of data points (between percentiles 25% and 75%, that is, between 5300 and 31700 data points) and; clients

---

[2]The implemented code is available in the following GitHub repository: github.com/OS-danielfranciscolopes/FL-Personalization

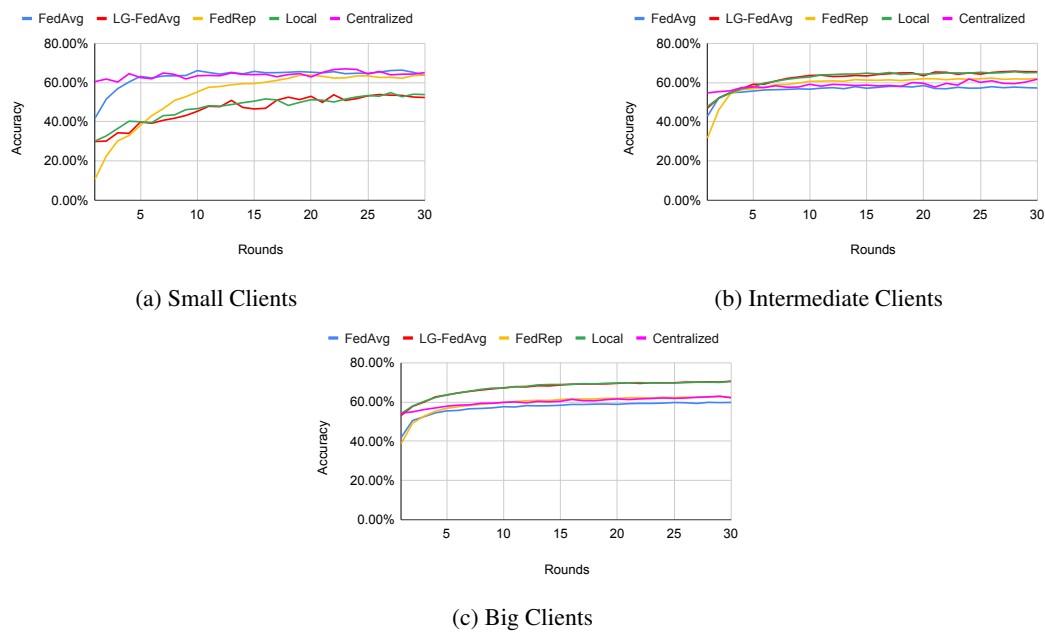

(a) Small Clients

(b) Intermediate Clients

(c) Big Clients

Figure 3: Accuracy of the various models for clients of different sizes.

with a big number of data points (above 75% percentile, that is, above 31700 data points). Figure 3 contains the graphs of the evolution of the accuracy by round and algorithm for each one of the client groups.

#### 4.2.1 Performance for Small Clients

Figure 3a shows the evolution of the average accuracy for the small clients throughout the training/communication rounds for each one of the models and algorithms. In terms of the federated models, we can see that the *FedAvg* algorithm is the one which obtains better accuracy, followed by the *FedRep* algorithm, meaning that personalizing the head is preferable to personalizing the body. The *LG-FedAvg* algorithm achieves the worst performance, a fact that can be justified by the few data points of the clients which do not allow for proper personalization of the body.

We can also observe that the centralized model is the one which achieves the highest accuracy (about 0.5% higher than *FedAvg*'s highest accuracy). However, its results are very similar to the ones of the *FedAvg* algorithm, even achieving worse accuracy than *FedAvg* in a considerable number of rounds. Lastly, we see that the local models are far inferior to both the centralized model and the *FedAvg* and *FedRep* algorithms, which shows the importance of client collaboration when clients have a small amount of data.

#### 4.2.2 Performance for Intermediate Clients

Figure 3b illustrates the evolution of the average test accuracy for the clients with an intermediate number of data points. We can see greater proximity between the accuracies of the three federated algorithms. Furthermore, the personalization algorithms are superior to *FedAvg*, in particular, the *LG-FedAvg* algorithm has the best accuracy, meaning that for clients with more data personalizing the body of the model is best since the greater amount of data allows for better personalization. Finally, *LG-FedAvg* is the only federated algorithm with similar accuracy to the local models (even achieving about 0,5% superior accuracy in some of the later rounds), and superior accuracy to the centralized model (about 4% to 6% in accuracy).

### 4.2.3 Performance for Big Clients

Figure 3c illustrates the evolution of the average accuracy for big clients. In this case, we can see a tendency similar to the one of the intermediate clients, where the personalization algorithms are superior to the *FedAvg* algorithm. Similarly, the *LG-FedAvg* algorithm is far superior to the *FedRep* algorithm and reaches an accuracy which is similar to the one of the local model (in some of the later rounds, *LG-FedAvg* is superior by $0.1\%$ to $0.2\%$ in accuracy) and superior to the centralized model (difference of $7\%$ to $8.5\%$).

### 4.3 Discussion

From the obtained results we can conclude that there is no strategy that is better than the others for all types of clients. For clients with few data points, collaboration on the full model is preferable, since the low amount of data makes personalization ineffective. The *FedAvg* algorithm, which trains the whole model collaboratively, obtains results very close to those of the centralized model, being superior in a considerable amount of rounds. Hence, it can be an alternative to the centralized model because it allows collaboration without sharing the clients' data (contrary to the centralized model).

As the number of data points increases (intermediate and big clients) the data becomes specific and in sufficient quantity to train, individual client models. Therefore, the centralized model becomes inferior to local models and personalization models are superior alternatives to the *FedAvg* algorithm. Also, the personalization of the body offers greater results than that of the head and actually, in some of the later rounds, slightly superior to local models by about $0.1\%$ to $0.2\%$ in accuracy, which indicates that the collaboration on the head might help these larger clients classify some data points which are more general and less specific to the client and that the local model fails to classify correctly. As such, we can conclude that for these clients, personalizing the representation is preferable to personalizing the classifier, which is somewhat surprising, since the literature mentions that it is expected for the heterogeneity to reside in the classifier and not in the representation.

In environments where data privacy is a requirement, the development of a hybrid approach between the *FedAvg* algorithm (for smaller clients) and the *LG-FedAvg* algorithm (for bigger clients) would allow bigger clients to collaborate in the construction of a federated model which would benefit the smaller clients without sharing their data and without needing to develop two models (one federated and one local), while also receiving a small boost in model performance.

## 5 Conclusions

The *Service Studio* platform developed by OUTSYSTEMS leverages GNNs in order to recommend possible next actions that the users might want to add to an action flow. In this paper, we presented the results of an experimental study, performed with the intent of assessing the possibility of substituting the current centralized model for federated algorithms which allow the creation of personalized models for each client, since the quantity of data of each client influences the performance of the algorithm. Therefore, the development of a mechanism which can combine the *FedAvg* algorithm (which reaches results very similar to the centralized model for smaller clients) with the *LG-FedAvg* algorithm (which by personalizing the body can reach similar results to the ones obtained using local models and greatly superior to the centralized model, for clients with more data) can allow for a good compromise between accuracy and data privacy. In the future, we intend to extend the study to more algorithms based on the parameter decoupling technique, possibly even considering algorithms where the model division is not restricted to only two parts (body and head). We also intend to test different personalization techniques, for instance, clustering; to evaluate the performance in a more complex problem with more possible classifications in order to see how each algorithm behaves with a more complex model and; to test how different hyperparameters influence the performance of each algorithm, such as the fraction of clients selected per communication round and the number of local training rounds.

## Acknowledgments and Disclosure of Funding

This work was performed in the scope of a curricular internship with OUTSYSTEMS and was partially funded from national funds, by FCT, Fundação para a Ciência e a Tecnologia, through the project UIDB/50021/2020.

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

# A Experimental Clients Statistics

Table 2: Statistics of the number of data points of the 33 selected clients

| min | max | mean | median | var | std-dev | Percentile | | | | |
|---|---|---|---|---|---|---|---|---|---|---|
| | | | | | | 25 | 75 | 90 | 95 | 99 |
| 80 | 374,860 | 57,855 | 10,175 | 9,006,809,135 | 94,904 | 1,003 | 77,790 | 182,641 | 266,686 | 346,235 |

# B Class Distribution

Table 3: Class distribution of the 33 selected clients.

| Class Name | Mean (%) | Std-dev (%) |
|---|---|---|
| Nodes.AjaxRefresh | 2.86 | 2.92 |
| Nodes.Assign | 18.33 | 4.56 |
| Nodes.AttachEmailContent | 0.01 | 0.03 |
| Nodes.Comment | 0.00 | 0.00 |
| Nodes.DataSet | 5.58 | 2.66 |
| Nodes.Download | 0.24 | 0.24 |
| Nodes.End | 22.93 | 3.18 |
| Nodes.ErrorHandler | 0.00 | 0.00 |
| Nodes.ExcelToRecordList | 0.28 | 0.46 |
| Nodes.ExecuteAction | 15.65 | 5.27 |
| Nodes.ForEach | 2.07 | 1.04 |
| Nodes.If | 10.77 | 2.28 |
| Nodes.JSONDeserialize | 0.13 | 0.18 |
| Nodes.JSONSerialize | 0.23 | 0.36 |
| Nodes.Outcome | 0.14 | 0.30 |
| Nodes.RaiseError | 0.77 | 1.36 |
| Nodes.RecordListToExcel | 0.13 | 0.16 |
| Nodes.RefreshQuery | 2.29 | 1.59 |
| Nodes.SendEmail | 0.10 | 0.15 |
| Nodes.Start | 0.00 | 0.00 |
| Nodes.Switch | 0.38 | 0.37 |
| Nodes.WebDestination | 3.64 | 2.86 |
| NRNodes.ExecuteClientAction | 6.14 | 7.02 |
| NRNodes.FeedbackMessage | 5.60 | 2.74 |
| NRNodes.JavascriptNode | 0.98 | 1.15 |
| NRNodes.TriggerEvent | 0.75 | 0.93 |

# C Experimental Model Parameters

Table 4: Experimental Model Parameters

| Parameter | Value |
|---|---|
| GNN type | FullGN |
| Input Dim [x, edge_attr, u] | [117, 18, 163] |
| GN Layer Output dim | 90 |
| Number of GNN Layers | 6 |
| Share Layers | Yes |
| Layer Normalization | Yes |
| Output Dim | 27 |
| Activation Function | ReLu |

# D    Experimental Hyperparameters

Table 5: Experimental Hyperparameters

| Model | Parameter | Value |
|---|---|---|
| **Centralized /** **Local** | Loss | Cross Entropy |
| | Batch Size | 128 |
| | Optimizer | SGD |
| | Learning Rate | 0.001 |
| | Training Epochs | 30 |
| **FedAvg /** **LG-FedAvg** | Loss | Cross Entropy |
| | Batch Size | 128 |
| | Optimizer | SGD |
| | Learning Rate | 0.001 |
| | Communication Rounds | 30 |
| | Local Training Epochs | 1 |
| **FedRep** | Loss | Cross Entropy |
| | Batch Size | 128 |
| | Optimizer | SGD |
| | Learning Rate | 0.001 |
| | Communication Rounds | 30 |
| | Head Training Epochs | 1 |
| | Body Training Epochs | 1 |

