# OpenReview forum: "Federated Learning for Predicting the Next Node in Action Flows"
_NeurIPS.cc/2022/Workshop/Federated_Learning — FL-NeurIPS 2022 Poster_

### Official Review · Reviewer_DDoS · 2022-10-17
**Personalization analysis for FL in Graph Neural Nets in the Cross-Silo setting**

This empirical applications paper applies federated learning to Graph Neural Network (GNN) models in the cross-silo context.  The work specifically considers the use of split model personalization techniques, comparing two (LG-FedAvg and FedRep) to three baselines (including non-personalized FL, e.g. FedAvg, and centralized learning), investigating the value of such personalization techniques in the context of clients with large variation in the degree of training data.

The work demonstrates that, for this application and for the particular training configuration considered, that personalization can be quite effective, but the effectiveness depends greatly on the amount of data the individual client has.  In particular, non-personalized FedAvg is highly effective for clients with little data, whereas for clients with more data, LG-FedAvg provides stronger results.  This invites future theoretical work constructing techniques that hybridize the use of FedAvg and LG-FedAvg to effectively serve all clients.

This work offers value in:
(1) novel domain/setting for FL (cross-silo + GNNs)
(2) empirical analysis suggesting need for future theoretical work

This work could be improved in the following ways:
(1) It is claimed that LG-FedAvg outperforms local learning for intermediate to large clients.  However, I don't believe it is actually possible to draw this conclusion from the presented data; in particular, the graphs in Figures 2b, 2c show the Local and LG-FedAvg lines essentially co-incididing, switching back and forth between which is outperforming the other.  In Checklist Item 3c, the authors assert that error bars were not included because "we are comparing previously tested algorithms" -- however, I suspect the inclusion of error bars in this work would show that there is no significant difference between LG-FedAvg and Local training in the settings considered.   If this is true, then it suggests that the appropriate hybridization might be the (simpler) hybridization of FedAvg and Local learning, with no need for personalized FL.

(2) In section 4.1, the authors assert that for FedRep, "one local training round for the head and one for the body were performed", and similarly "a single local training round" for FedAvg and LG-FedAvg.  It is unclear whether the results were sensitive to these choices, nor why these choices would be obvious. Do they lead to under-personalization?  Over-personalization?  Are somehow robust to both?

---

> ### Author Response · Authors · 2022-11-29
> **Response to Reviewer DDoS**
>
> The authors would like to thank the reviewer for the insightful comments and suggestions.
>
> - The hypothesis expressed in comment (1) is interesting and we aim at performing additional experiments in the future to confirm it.
>
> > In section 4.1, the authors assert that for FedRep, "one local training round for the head and one for the body were performed", and similarly "a single local training round" for FedAvg and LG-FedAvg. It is unclear whether the results were sensitive to these choices, nor why these choices would be obvious.
>
> - We have added clarification stating that the chosen FL hyperparameters were based on the ones used in the LG-FedAvg and FedRep papers. As future work, we intend to test how varying the number of local computation rounds influences the performance of the algorithms.

---

### Official Review · Reviewer_xMqC · 2022-10-18
**Lacks in the novelty and experimentation but recommend acceptance due to application on a new domain**

The authors provide a comparison of two personalized algorithms (LG-FedAvg and FedRep) on a proprietary task involving action flows based on GNN models. The main contribution of the paper is to study these algorithms in a novel problem setup based on GNN models.
The paper is very limited in the details it provides (I guess due to the blind review process), but the work is a step towards studying FL for other domains than just image and NLP.  For this alone, I believe the paper should be accepted.

However, the work in itself is severely limited in the content and novelty. Although the only contribution is the experimentation, the current results are not adequate in providing a clear picture. For example, the following could be helpful:
- an ablation analysis on the number of clients available in the federation,
- a comparison with clustering based FL approaches [2,3] (since it is another popular approach for personalization).
- Discussion on what was the basis for choosing the two algorithms? Why not other personalized algorithms? Why not other general FL algorithms?

I feel the clustering technique could alleviate the issue observed in their experiments, since the large clients can train locally and the small clients can form clusters and train together. There could be a one-way transfer of knowledge from the large clients to the smaller clients' clusters which eventually gets distilled into the workers.

Moreover, the details provided in the experimentation section are confusing:
- "Since the dataset is balanced": There was no prior mention about the balance/imbalance in the dataset. Also, balanced with respect to the number of labels/actions at each worker, or the number of data points at each worker?
- "From this dataset we selected 33 clients for evaluation" : Are these clients used only for the final evaluation, or for training of the algorithm as well? (in line 156, the authors mention that all 33 clients participate at every round).
- How was the hyper parameter tuning done?

Instead of providing these details, the authors spend almost 2 pages on explaining about FL or past algorithms which could have been avoided. The literature review also feels inadequate. For example, the authors could have considered/compared personalization techniques from [1] which extends the setup to beyond the body-head architecture and from clustering techniques [2,3].

I sincerely hope that the authors try to improve the work along these lines. I hope that since the task is novel, the authors will provide more details about the dataset/task upon acceptance. The authors mention that the code will be made available, but no mention of the dataset. Would that be made available too?

[1] Pillutla et al. (ICML 2022) Federated Learning with Partial Model Personalization

[2] Sattler, Felix, Klaus-Robert Müller, and Wojciech Samek. "Clustered federated learning: Model-agnostic distributed multitask optimization under privacy constraints." IEEE transactions on neural networks and learning systems 32.8 (2020): 3710-3722.

[3] Briggs, Christopher, Zhong Fan, and Peter Andras. "Federated learning with hierarchical clustering of local updates to improve training on non-IID data." 2020 International Joint Conference on Neural Networks (IJCNN). IEEE, 2020.

---

> ### Author Response · Authors · 2022-11-29
> **Response to Reviewer xMqC**
>
> The authors would like to thank the reviewer for the insightful comments and suggestions.
>
> > an ablation analysis on the number of clients available in the federation
>
> - In this study, we did not consider varying the fraction of participating clients per communication round. As future work, we intend to perform a more in-depth study where we vary this hyperparameter.
>
> > a comparison with clustering based FL approaches [2,3] (since it is another popular approach for personalization).
>
> - This paper addresses the parameter decoupling technique. As future work, we intend to experiment with other personalization techniques. The clustering technique referred in the review is definitely worth assessing in this context.
>
> > There was no prior mention about the balance/imbalance in the dataset. Also, balanced with respect to the number of labels/actions at each worker, or the number of data points at each worker?
>
> - The data set is balanced in terms of the labels at each worker, we have added the class distribution in Appendix B. As for the number of data points of each worker it is highly unbalanced as it can be seen by the statistics of the number of data points for the 33 clients present in Appendix A.
>
> > Are these clients used only for the final evaluation, or for training of the algorithm as well? (in line 156, the authors mention that all 33 clients participate at every round).
>
> - We have added to the text the exact number of clients used in both training and testing.
>
> > How was the hyper parameter tuning done?
>
> - We have added clarification in the paper stating that the model hyperparameters were obtained in a previous empirical evaluation. As for the FL hyperparameters, we have clarified that these were based on the experimental evaluation of both LG-FedAvg and FedRep.
>
> > The authors mention that the code will be made available, but no mention of the dataset. Would that be made available too?
>
> - The data set includes confidential client data and cannot be made available. The source code has been made available and a link to the repository has been included in the paper.

---

### Official Review · Reviewer_hhE6 · 2022-10-19
**Interesting experience sharing, but novelty and details can be further improved**

## Strengths:
* The paper provides an interesting experience sharing for FL with GNNs with a real-world dataset.
* The research artifacts could facilitate further research.
## Weaknesses:
* The key observation is not particularly novel. It is well known that personalized FL and local models typically do not work well for clients with limited data. This was one of the main motivations in the FedRep paper.
* Personalized FL is a well-studied problem with various objectives (accuracy, fairness, robustness, etc.). The paper will benefit from clearly stating the objective, as well as using more personalized FL baselines, such as model interpolation (e.g., APFL), meta-learning (e.g., Per-FedAvg), or personalized weighted combination (e.g., FedFomo).
* The paper can provide more details, such as (1) train/test splits, (2) how the hyperparameters were determined, (3) is there any heterogeneity in label/feature distribution among different clients, and (4) how the 33 clients were selected.

---

> ### Author Response · Authors · 2022-11-29
> **Response to Reviewer hhE6**
>
> The authors would like to thank the reviewer for the insightful comments and suggestions.
>
> > The key observation is not particularly novel. It is well known that personalized FL and local models typically do not work well for clients with limited data. This was one of the main motivations in the FedRep paper.
>
> - The reviewer is right in that previous research has shown that, in general, local models and personalized FL models have limited performance when the clients have small amounts of data. However, we were interested in assessing the practical implications of this effect for our concrete use case.
>
> > The paper will benefit from clearly stating the objective, as well as using more personalized FL baselines, such as model interpolation (e.g., APFL), meta-learning (e.g., Per-FedAvg), or personalized weighted combination (e.g., FedFomo).
>
> - We have rephrased the text to make clear that our goal is to develop FL models that maintain or improve the performance (measured in accuracy) in relation to local and centralized models while ensuring data privacy.
>
> - In this paper we only focused on the parameter decoupling techniques. We intend to explore other personalization techniques in the future.
>
> > The paper can provide more details, such as (1) train/test splits, (2) how the hyperparameters were determined, (3) is there any heterogeneity in label/feature distribution among different clients, and (4) how the 33 clients were selected.
>
> -   We have added information regarding the train and test splits to the first paragraph of the Experimental Setup section (Section 5.1).
>
> - The model hyperparameters were obtained empirically by OutSystems previously to this work,  and we have reused the values in our tests. As for the FL training parameters we have added that these were chosen considering the FL hyperparameters used in both the LG-FedAvg and FedRep papers.
>
> - We have added the class distribution in Appendix B.
>
> - The process of client selection is now described in the second paragraph of Section 5.1.

---

### Decision · Program_Chairs · 2022-10-20

Accept (Poster)